# The Presence of Two *MyoD* Genes in a Subset of Acanthopterygii Fish Is Associated with a Polyserine Insert in MyoD1

**DOI:** 10.3390/jdb11020019

**Published:** 2023-04-28

**Authors:** Lewis J. White, Alexander J. Russell, Alastair R. Pizzey, Kanchon K. Dasmahapatra, Mary E. Pownall

**Affiliations:** Biology Department, University of York, York YO10 5DD, UK

**Keywords:** myogenesis, transcriptional regulators, amino acid repeat proteins, intrinsically disordered regions

## Abstract

The *MyoD* gene was duplicated during the teleost whole genome duplication and, while a second *MyoD* gene (*MyoD2*) was subsequently lost from the genomes of some lineages (including zebrafish), many fish lineages (including *Alcolapia* species) have retained both *MyoD* paralogues. Here we reveal the expression patterns of the two *MyoD* genes in *Oreochromis* (*Alcolapia) alcalica* using in situ hybridisation. We report our analysis of MyoD1 and MyoD2 protein sequences from 54 teleost species, and show that *O. alcalica*, along with some other teleosts, include a polyserine repeat between the amino terminal transactivation domains (TAD) and the cysteine-histidine rich region (H/C) in MyoD1. The evolutionary history of *MyoD1* and *MyoD2* is compared to the presence of this polyserine region using phylogenetics, and its functional relevance is tested using overexpression in a heterologous system to investigate subcellular localisation, stability, and activity of MyoD proteins that include and do not include the polyserine region.

## 1. Introduction

MyoD is an important developmental switch that determines cell fate by driving skeletal muscle cell differentiation. *MyoD* is part of a family of myogenic regulatory genes coding for basic helix-loop-helix (bHLH) transcription factors that activate skeletal muscle specific gene expression during development and muscle regeneration [1,2,3]. *MyoD* is highly conserved, with orthologous genes across all vertebrates as well as related regulators in invertebrates [4,5]. One copy of the *MyoD* gene is present in the birds and mammals that have been investigated, and the two *MyoD* alleles reported in allotetraploid *Xenopus laevis* represent its homoeologous subgenomes [6], while only one copy is present in the diploid *Xenopus tropicalis* [7]. In contrast to terrestrial vertebrates, many fish species have more than one *MyoD* gene [8,9].

A whole genome duplication (WGD) event in teleosts (around 350 million years ago) means that there are often two fish paralogues of a gene that is represented only once in other vertebrate lineages [10]. This is true for *MyoD,* which was duplicated during the teleost WGD and subsequently lost in some lineages, such as the Ostariophysi that includes zebrafish (*Danio rerio*) [8]. Two non-allelic *MyoD* genes were first identified in sea bream [11], and later in halibut [12], where differential gene expression of *MyoD1* and *MyoD2* were described in both species. The Protacanthopterygii also lost the *MyoD2* gene, however lineage specific duplication (LSD) of *MyoD1* in salmon (and some other fish) has resulted in multiple *MyoD* genes. The phylogenetic and syntenic analyses support the nomenclature of these genes as *MyoD1a*, *1b*, *1c* [8] as these *MyoD1* genes result from LSD and form a monophyletic clade together with the single *MyoD* present in terrestrial vertebrates. Some well-studied Acanthopterygii, such as sticklebacks (*Gasterosteus aculeatus*), medaka (*Oryzias latipes*), fugu (*Takifugu rubribes*), and cichlids are among the fishes that retain both *MyoD1* and *MyoD2* [12]; this includes the extremophile Alcolapia cichlid species of East Africa.

African cichlids are distributed across a variety of habitats, from benign to extreme, and the *Alcolapia* species are a unique radiation of cichlids that have evolved rapidly from freshwater ancestors to inhabit one of the most extreme aquatic environments supporting fish life. *Alcolapia* are endemic to Lakes Natron and Magadi, where the water temperature, salt concentration and alkalinity are very high and the level of dissolved oxygen fluctuates [13]. *Alcolapia* is the only genus of vertebrate known to survive in these lakes and their species include *O. alcalica*, *A. grahami*, *A. latilabris* and *A. ndalalani* [14]. The development of *O. alcalica* has been recently described in [15] and their embryos provide an opportunity to investigate developmental regulators including MyoD in a non-model organism with unique adaptations to its extreme environment.

Our analysis of *MyoD1* and *MyoD2* in *O. alcalica* includes a phylogeny that suggests that the retention of two *MyoD* paralogues is associated with a polyserine insertion amino terminal to the bHLH domain of the MyoD1 protein. We analyse the expression pattern of *MyoD1* and *MyoD2* in *O. alcalica* embryos, and find a notably higher expression of *MyoD1*. We investigate whether the polyserine insert is associated with protein stability, as MyoD has been shown to be an inherently unstable protein [16,17,18] where multi-site phosphorylation of serines is known to contribute to the regulation of its stability [16]. Amino acid repeats in transcription factors have recently been found to influence interactions with transcriptional co-factors [19]; we look at the possibility that novel protein-protein interactions could change the activity or subcellular localisation of *O.alcalica* MyoD1. The significance of the polyserine insert in a subset of ray-finned fishes (the Acanthopterygii), which evolved alongside the retention of *MyoD2*, is discussed, and the possibility of divergent functions of MyoD1 is considered in the context of current understanding of intrinsically disordered regions in transcription factors.

## 2. Methods

### 2.1. Sequence Analysis and Phylogeny of MyoD Genes in Teleosts

*Oreochromis niloticus MyoD1* and *MyoD2* sequences were aligned to the unpublished *Oreochromis (Alcolapia) alcalica* whole genome illumina sequence (Dashamapatra, in preparation) so as to identify and extract the coding sequences of these myogenic regulators in *O. alcalica*. The published phylogeny of ray-finned fish Actinopterygii [20] was used to construct a best accepted species phylogeny for all fish species with sequence data available for *MyoD* genes and whose position in the mass teleost phylogeny could be inferred. The R packages phytools, ape, and geiger [21,22,23] were used to prune the phylogeny so only these species were included in the final tree. The final tree contained 43 fish species and outgroups of a tetrapod (the frog, *Xenopus tropicalis*) and a coelacanth (*Latimeria chalumnae*). The presence of the different *MyoD* genes and presence or absence of the polyserine region were compared to the species phylogeny to infer evolutionary history.

A separate amino acid sequence alignment containing 97 genes from 54 teleost species was constructed. All sequences were identified and downloaded from ENSEMBL (accession numbers are in the Appendix A). This phylogeny was produced by aligning amino acid sequences using MEGAX [24] modelled using Jones-Taylor-Thornton with frequency and gamma distribution, as it was determined to be the best fit for the data based on the programme’s model testing software. Bootstrap analysis using 100 repeats was used to evaluate the resulting phylogeny.

### 2.2. Cloning and Expression Analysis

Reverse transcription polymerase chain reaction (RT-PCR) was used to isolate cDNA clones from *A. alcalica*. RNA was extracted from day 5 embryos using TriReagent (Sigma-Aldrich, Steinheim, Germany) according to the manufacturers’ guidelines. For cDNA synthesis, 1 µg of total RNA was reverse transcribed with random hexamers (Thermo Scientific, Waltham, MA, USA) and superscript IV (Invitrogen, Waltham, MA, USA). PCR was carried out using Promega PCR master mix and the primers listed in Appendix A. For overexpression studies, HA-tagged full-length products were amplified using the same approach but with the *Pfu* polymerase (Promega, Madison, WI, USA). The full-length product representing the open reading frame coding for each protein were cloned into pCS2+ and the coding sequences for *O. alcalica* MyoD1 and MyoD2 were submitted to NCBI (accession numbers: MW448164, MW448165). mRNA was synthesised in vitro using SP6 mMESSAGE mMACHINE^®^ (Invitrogen) according to manufacturer’s guidelines. For gene expression analysis, PCR products were directly cloned into pGEM T-easy and antisense probes for *O. alcalica* MyoD1 and MyoD2 were generated by linearising and using in vitro run-off transcription to incorporate a DIG labelled UTP analogue.

To determine the expression pattern of *MyoD1* and *MyoD2* in *O. alcalica*, embryos were collected at 2 and 4 days of development, fixed for 1 h in MEMFA (0.1 M MOPS pH 7.4, 2 mM EGTA, 1 mM MgSO4, 3.7%formaldehyde) at room temperature and stored at −20 °C in 100% methanol. Embryos were rehydrated and treated with 10 μg/mL proteinase K at room temperature. After post fixation and a 2-h pre-hybridisation, embryos were hybridised with the probe at 68 °C in hybridisation buffer (50% formamide (Ambion, Austin, TX, USA), 1 mg/mL total yeast RNA, 6 × SSC, 100 µg/mL heparin, 1 × Denharts, 0.1% Tween-20, 0.1% CHAPS, 10 mM EDTA). Embryos were extensively washed at 68 °C in 2 × SSC + 0.1% Tween-20, 0.2 × SSC + 0.1% Tween-20 and brought to room temperature in Maleic acid buffer (MAB; 100 mM maleic acid, 150 mM NaCl, 0.1% Tween-20, pH7.8). This was replaced with blocking buffer (1 × MAB, 10% Blocking reagent (Roche, Krackeler Scientific, Albany, NY, USA), 20% heat-treated lamb serum) for 2 h. Embryos were incubated overnight (rolling at 4 °C) with fresh pre-incubation buffer and 1/2000 dilution of anti-DIG coupled with alkaline phosphatase (Roche), and expression was visualised using the substrate BM purple (Roche).

### 2.3. Protein Expression

For functional analyses, heterologous protein expression was undertaken using *Xenopus laevis* embryos. 4 ng of synthetic mRNA coding for each of the proteins *O. alcalica* MyoD1, and *D. rerio* MyoD1 was injected into both blastomeres of two-cell embryos. After culturing for 5 h at 20 °C the embryos were at the blastula stage, and some were fixed in MEMFA and stored at −20 °C in methanol for later analysis by immunofluorescence. The rest of the embryos were used for collecting explants for activity and stability analysis. The cells from the animal hemisphere were explanted and cultured in normal amphibian media (NAM) until the control sibling embryos reached Nieuwkoop and Farber stage 14 (early neurula). These samples were collected for qPCR. Another set of explants were cultured in NAM + 10 ug/mL cycloheximide to inhibit protein synthesis and collected by snap freezing on dry ice at three timepoints: prior to treatment (T0), at 1 h post treatment (T1) and at 3 h post treatment (T2). The samples were directly homogenised in 50 uL of 2× Laemmli sample buffer and heated to 80 °C for 5 min before analysing by SDS-PAGE (Sodium dodecyl-sulfate polyacrylamide gel electrophoresis)and Western blotting.

### 2.4. Western Blotting

To test for protein stability over time, cyclohexamide-treated *X. laevis* explants were boiled in sample buffer, microfuged briefly, and proteins were separated by SDS-PAGE before transferring onto a PVDF (polyvinylidene fluoride) membrane by electroblotting. Membranes were blocked in 5% Marvel milk powder/PBST (phosphate buffered saline with Triton-X 100) for 1 h before incubating with the primary antibody of anti-HA (Sigma) at a concentration of 1 in 4000 overnight at 4 °C. After extensive washing, the membrane was incubated in a secondary antibody (1 in 4000 goat anti-mouse conjugated with HRP (Invitrogen) for an hour at room temperature. Membranes were washed and visualised using a BM Chemiluminescence kit (Roche) and hyperfilm ECL. After stripping in 50 mM Tris-HCl (pH 7) with 2% sodium dodecyl sulfate (SDS) and 50 mM dithiothreitol (DTT), the antibody β-tubulin (Cell Signalling Technology), at a concentration of 1 in 10,000, followed by a goat anti-rabbit secondary (Invitrogen) at a concentration of 1 in 2000, provided a normalisation control using the same method.

### 2.5. Immunofluorescence

Immunofluorescence was carried out on whole *X. laevis* embryos overexpressing with HA-tagged MyoD proteins according to Christen and Slack, 1999, [25]. Embryos fixed in MEMFA and stored in methanol at −20 °C were gradually rehydrated and then permeabilised in potassium dichromate treatment followed by 5% H_2_O_2_ in PBS. After washes and an hour blocking at room temperature, embryos were incubated overnight at 4 °C in BBT (PBS + 1%BSA, 0.1% Triton X-100) and 5% horse serum with a 1:1000 concentration anti-HA antibody (Sigma). The secondary anti-mouse IgG Alexa Fluor 488 (Abcam, Cambridge, UK) was used at 1:1000 and embryos were incubated for 1 h at room temperature.

### 2.6. qPCR

Primers for qPCR were designed using Integrated DNA Technologies PrimerQuest tool. Products were designed to be between 75 and 110 base pairs in length and spanning exon junctions (a table listing all primers is provided in the Appendix A). qPCR was performed using the QuantStudio 3 (Admiral; A28567) and analysis of results used the Livak method (Livak and Schmittgen, 2001, [26]) of 2^−ΔΔCT^. All statistical tests were performed on the untransformed ΔCT values. Average CT was normalised to the housekeeping gene *Dicer* and pairwise *t*-tests were carried out comparing the mean relative expression for control and experimental sets (siblings) for each gene. Three technical repeats as well as three biological replicates were analysed.

## 3. Results

*MyoD1* and *MyoD2* genes were cloned from *Oreochromis (Alcolapia) alcalica* and sequenced. The protein sequence of *O. alcalica* MyoD1 was compared to MyoD1 sequences reported for tetrapods and other teleost species. A conspicuous amino acid repeat comprising 16 Serine and 2 Proline residues is evident in *O. alcalica* MyoD1 and similar polyserine inserts are found in the same region of MyoD1 in Fugu and the Amazon Molly, while it is not present in MyoD1 in zebrafish, the rainbow trout or the common carp (Figure 1). The polyserine region is located amino-terminal to the bHLH DNA binding domain, between the transactivation domains (TAD) and the cysteine-histidine rich region (H/C) of MyoD1; these regions are essential for the biochemical activity as transcription factors and are highly conserved across all vertebrates (See Appendix A). On further investigation, a polyserine insertion was found to be present in a large number of teleost *MyoD1* genes analysed (see Appendix A for an alignment of this region from 32 MyoD1 species that include this region). The insertion ranges from 16 to 27 amino acids that are mostly serine, but also includes proline and leucine residues, with some heterogeneity between species. The existence of this domain has been described previously [27], but no study has investigated its prevalence or relevance.

### 3.1. Phylogenetic Analysis Reveals Relationship among MyoD Genes in Teleost Fish

A mass alignment of the amino acid sequences of multiple *MyoD* genes found in the genomes of teleost fish was produced and compared via phylogenetics. The analysis produced two separate clusters with strong bootstrap support indicating that the multiple genes in these fish are comprised of *MyoD1* and *MyoD2* (Appendix A), where some species have multiple *MyoD1* genes. MyoD1 sequences with and without the region coding for a polyserine insertion cluster together as MyoD1 in this analysis. The phylogenetic tree shown in the Appendix A should not be used to infer species relatedness as it is derived from the alignment of a single protein.

The teleost phylogeny published by Hughes et al. [20] was used to investigate the evolution of the different *MyoD* genes in these fish species (Figure 2). Comparison of this tree with the sequence data for *MyoD1* and *MyoD2* in teleost fish revealed clades where the polyserine insertion in MyoD1 had occurred and identified which clades had both a *MyoD1* and *MyoD2* gene. This analysis indicates that the inclusion of a polyserine insertion in MyoD1 is found in species that retain a *MyoD2* gene (Figure 2, red lines). This suggests that the polyserine insertion arose as a single evolutionary event and was incorporated and retained by species in this lineage. It has been well described that some fish species (which have neither a polyserine insertion in their MyoD1 nor a *MyoD2* gene), such as salmon and cod [28,29,30], have multiple *MyoD1* genes arising from single lineage duplication events (Figure 2, blue lines).

### 3.2. Expression Patterns of MyoD1 and MyoD2 in Developing O. alcalica Embryos

In situ hybridisation using DIG labelled cRNA probes for the two *MyoD* genes was used to determine the spatial expression pattern of *MyoD1* and *MyoD2* in early *O. alcalica* development (Figure 3). In the two stages examined in this study the staining for *MyoD1* expression was found to be consistently stronger compared to *MyoD2*, suggesting that *MyoD1* may have a higher level of transcription in the developing embryo; however, this would need to be confirmed using a quantitative assy such as RT-qPCR. The spatial expression is similar; both genes show expression in the developing somites along the body axis (black arrows), in the pectoral fin buds (white arrows) and in facial muscle tissue (white arrowheads); this is typical of *MyoD* expression in other teleosts [31]. Expression of *MyoD1* and *MyoD2* in day 3 embryos is included in the Supplementary data Appendix A. Early expression of *MyoD1* (at 2dpf) is detected in what might be adaxial cells (arrowhead, Figure 3A, also see a flat mount in Appendix A); this expression was not detected in embryos analysed for *MyoD2* expression.

### 3.3. O. alcalica and D. rerio MyoD1 Proteins Show Nuclear Localisation

The possibility that the polyserine insert in *O. alcalica* MyoD1 could influence its ability to move to the nucleus was tested by expressing tagged proteins in cells of a heterologous system (*Xenopus*). mRNA coding for HA-tagged MyoD proteins representing *MyoD1* from both *O. alcalica* and *D. rerio* was injected into *Xenopus* embryos that are well known to rapidly and efficiently translate injected mRNAs into protein. Immunostaining was used to detect the subcellular localisation of the proteins (Figure 4). Images show that MyoD1 proteins from both *O. alcalica* and *D. rerio* are localised to the nucleus of the cells of the blastula stage *Xenopus* embryos, as expected for a transcription factor.

### 3.4. O. alcalica MyoD1 Protein Persists Longer Than D. rerio MyoD1

MyoD is known to be inherently unstable, where multiple phosphorylation sites contribute to protein turnover or perdurance [32]. The polyserine insert includes some putative consensus sites for proline directed kinases, therefore, we used the *Xenopus* system to express MyoD1 proteins and compare their stability over time. To compare the stability of the MyoD1 protein that includes a polyserine insertion found in *O. alcalica* to MyoD1 protein lacking this insert found in *D. rerio*, mRNAs coding for HA-tagged proteins were injected into *Xenopus laevis* embryos at the one cell stage. After culturing a few hours to allow for protein expression, tissue explants were dissected and incubated in cyclohexamide, a translational inhibitor, to prevent any further protein expression. The explants were collected at three time points and the proteins were detected using Western blotting (Figure 5). Both *D.rerio* and *O.alcalica* MyoD are robustly expressed at time zero (T0) and after an hour (T1). After three hours (T2), *O. alcalica* MyoD1 protein was consistently still detectable whereas, at the same time point, *D. rerio* protein was absent. The degradation of the *Danio* MyoD1-HA in this assay is comparable to the turnover of mouse MyoD1 seen in other studies using a similar approach [33]. These data suggest that *O. alcalica* MyoD1 is more stable than *D. rerio* MyoD1, a feature that could be due to the polyserine insertion.

### 3.5. Transcriptional Activity

To test the transcriptional activity of the MyoD1 proteins, heterologous expression in *Xenopus* was carried out as described above. These explants provide a naïve, pluripotent cell population capable of responding to MyoD by activating the expression of skeletal muscle specific genes (Gurdon 1990, [34]). qPCR analysis of *myosin heavy chain 4* (myh4) and *alpha actin 3* (act3) transcription in response to expression of MyoD1 proteins is shown in Figure 6. Equivalent levels of the two proteins are expressed in *Xenopus* explants (as determined by Western blotting and immunofluorescence); however, *Danio* MyoD1 has more activity. The transcriptional activation of the *myh4* (Figure 6A) and *act3* (Figure 6B) genes is significantly higher in response to *D.rerio* MyoD1 compared to *O.alcalica* MyoD1 in this assay. It is possible that the proximity of the polyserine insertion to the transcriptional activation domain (TAD) of *O.alcalica* MyoD1 could impact its strength as a transcription factor.

## 4. Discussion

Here we report the correlation between the presence of two MyoD genes and the inclusion of a polyserine insert in MyoD1 in a large number of Acanthopterygii fish species. The polyserine insert is a simple sequence repeat region that can arise from DNA slippage during replication and which, once established, can facilitate adaptive changes [35,36]. Our observation that the polyserine insert in MyoD1 is only found in genomes that retain a second MyoD gene is consistent with a study by Radó-Trilla et al. [37] demonstrating that the rate of appearance of low complexity amino acid repeat regions is increased in duplicated genes. Tandem repeats of amino acids result in intrinsically disordered regions (IDRs), and when present adjacent to the functional domain of a protein (in this case the DNA binding domain of a transcription factor), can be a source of functional change, for instance, by allowing it to interact with other proteins (for instance [38]). IDRs are known to interact with modulators of chromatin such as MED1 and can establish liquid-like condensates that concentrate super enhancers (see [39]). The proximity of the polyserine region to the bHLH domain in the *O. alcalica* MyoD1 protein also provides a regional template for modification by phosphorylation that could provide another mechanism to regulate interactions with protein partners and/or enhancers ultimately influencing its activity as a transcription factor. Our studies do not address the possibility of this level of regulation of MyoD activity, but we have some evidence that suggest that the polyserine insert is associated with a more stable protein.

### 4.1. Phylogenetic Analysis Reveals a Single Evolutionary Event for Inclusion of the Poly-Serine Domain in MyoD1

All teleost genomes derive from the teleost specific whole genome duplication, an event that is thought to underpin the remarkable diversity of extant fishes [9]. Although recent fossil evidence suggests the timeline of species diversification is significantly removed from the WGD event [40], and the return to the diploid state is very slow, such that paralogous genes evolve long after the WGD [41,42]; nonetheless WGD is an important driver of diversity in teleosts. Analysis of the amino acid sequences of MyoD1 and MyoD2 from a broad range of teleost species confirms that a large number of fish retain two separate MyoD genes. The presence of two distinct *MyoD* genes in some species like *O. alcalica* is different from the multiple *MyoD1* genes found in the genomes of a few species of fish such as those in the Salmonidae which arose from a lineage specific genome duplication [30]. Our investigation into the evolution of MyoD2, using the teleost phylogeny (Figure 2) [20], could suggest a single evolutionary event not encompassing all teleosts; however, it has been demonstrated that teleost species which lack a *MyoD2* gene retain the ghost loci in the same region [8,9]. This provides evidence for the whole genome duplication event resulting in a second *MyoD* gene in teleosts, *MyoD2*, which was subsequently lost in a number of lineages. Interestingly, the presence of a region coding for a poly-serine domain in the sequence of *MyoD1* is strongly linked to the presence of *MyoD2*. The phylogenetic analysis comparing the presence and absence of a poly-serine insertion in MyoD1 suggests that the region had a single evolutionary event and was retained. The observation that its presence is closely tied to the persistence of MyoD2 in a large subset of the Actinopterygii suggests an evolutionary advantage and is consistent with the ideas presented by Radó-Trilla et al. in their 2015 study [37].

### 4.2. Preliminary Functional Analysis of MyoD1 Polyserine Region and Future Directions

The occurrence of a single amino acid repeat region in protein sequences is common in animals and plants that are a result of ‘slippage’ during chromosome replication: while the DNA strands are denatured, the displacement of complementary strands results in mispairing and the incorporation of additional bases. These are then assimilated during recombination; in this way a growing string of repeat amino acids may emerge [43,44,45,46]. The presence of repeat regions from replication slippage has led to the hypothesis that repeat amino acids are more likely to occur in proteins with relaxed selection pressures, such as duplicated proteins [47]. Repeat regions within proteins present new genetic material for evolutionary change and have been associated with novel protein function, such as increased protein flexibility or stability, as well as being implicated in a number of diseases [48,49]. Although the repeat amino acid region found in teleost MyoD1 is a somewhat variable across species, being made up of at least eleven serine residues; the variability is likely a consequence of the increased mutation rates which occur in short repeat regions of DNA [36,47]. The presence of two or more serine-proline (SP) sequences (see Appendix A) suggests possible regulation by proline directed kinases such as cdk2 that is known to regulate MyoD stability during the cell cycle [50].

Our analyses have not resulted in conclusions regarding any evolutionary advantage to retaining the polyserine region of MyoD1. We expressed MyoD1 protein from a species with and a species without the insertion, *O. alcalica* and *D. rerio* respectively, and analysed subcellular localisation, stability and activity. We found *D. rerio* MyoD1 to behave as reported for mouse MyoD1: a labile, nuclear protein that strongly activates the transcription of skeletal muscle specific genes ([32,51,52,53]. Analysis by western blot showed that the polyserine containing MyoD1 from *O. alcalica* was more stable; present for at least three hours, while *D. rerio* MyoD1 was no longer detected by this time point. Like *D. rerio,* mouse MyoD1 does not contain a polyserine region and has also been shown to be unstable with a half-life of about 45 min [33]. Further analyses, using chimeric MyoD1 proteins where the polyserine region is removed from the *O. alcalica* sequence and inserted into the *D. rerio* sequence, would be a useful way to determine whether this region is responsible for the difference in stability. Despite the apparent increase in stability, *O. alcalica* MyoD1 did not exhibit any enhanced ability to activate transcription, at least not of the two contractile protein genes examined here. Both *D. rerio* and *O. alcalica* MyoD1 proteins were able to activate robust expression of skeletal muscle actin (*act3*) and myosin (*myh4*) genes in the *Xenopus* explant assay (first used by [34]).

### 4.3. Differences in the Expression Patterns of MyoD1 and MyoD2 in O. alcalica

Expression analysis of *MyoD1* and *MyoD2* in the Atlantic halibut (*Hippoglossus hippoglossus*) and gilthead seabream (*Sparus aurata*) demonstrated that *MyoD2* is expressed in a subset of somites and absent from adaxial cells [11,12]. The Atlantic halibut, a flatfish, showed further differential expression of the *MyoD* isoforms with left-right asymmetry of *MyoD2* expression [12]. Our findings also describe some differences in expression of the two *MyoD* genes in the developing embryos of *O. alcalica.* Overall, *MyoD2* is expressed at a much lower level, however both genes show strongest expression in somites along the body axis, developing pectoral fin buds and facial muscle tissues, as would be expected for genes involved in muscle cell determination and differentiation. *MyoD1* was found to be more strongly expressed in somites, facial muscle and pectoral fin buds than *MyoD2;* in addition, *MyoD1* and not *MyoD2* is expressed in the adaxial cells, which gives rise to slow muscle tissue [54]. One could speculate that *MyoD1* expression in somites and adaxial cells suggests it is important in both fast and slow muscle development, while *MyoD2* may only play a role in fast muscle development.

## 5. Conclusions

Two *MyoD* genes are found in the genomes of a large number of teleost fish, products of the teleost specific whole genome duplication event, and the duplicate was subsequently lost in some lineages. The MyoD1 protein in those lineages which retained a MyoD2 have evolved a polyserine region between the transactivation domains (TAD) and the cysteine-histidine rich region (H/C). This region has been retained, at least by the species included in this study, which would suggest a functional role. Preliminary analysis suggests it may increase protein stability; however, this would need to be interrogated with further research. Finally, consistent with findings in other fish species with both *MyoD* genes, the expression patterns of *MyoD1* and *MyoD2* in developing embryos of *O. alcalica* show both overlapping and distinct expression, suggesting the possibility of some subfunctionalization of the two proteins.

## Figures and Tables

**Figure 1 jdb-11-00019-f001:**
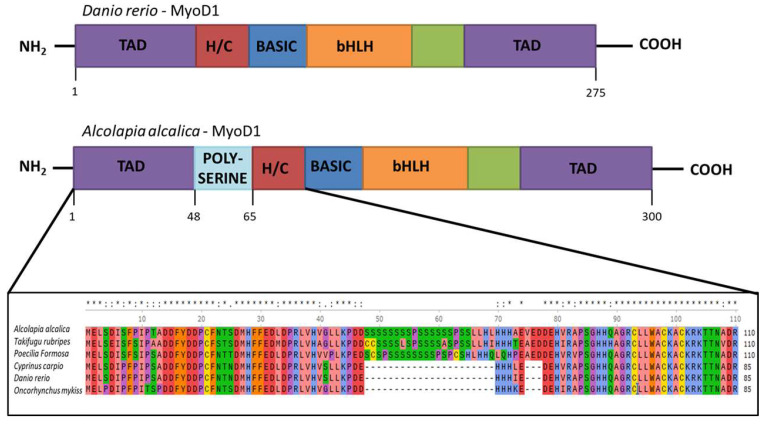
Inclusion of a polyserine insertion in MyoD1 in some fish species. Comparison of the protein structure of MyoD1 from *Danio rerio* (**above**) and *Oreochromis (Alcolapia) alcalica* (**below**). The enlargement of the schematic shows an alignment of the amino acid sequences from the indicated part of MyoD1 proteins from *Oreochromis (Alcolapia) alcalica*, *Takifugu ruberipes*, *Poecilia formosa*, *Cyprinus carpio*, *Danio rerio, Oncorhynchus mykiss* including the polyserine insertion in some species. Multiple species alignment of MyoD1 with and without the polyserine insertion. * represents exact match and: represents residues where the majority of species show conservation.

**Figure 2 jdb-11-00019-f002:**
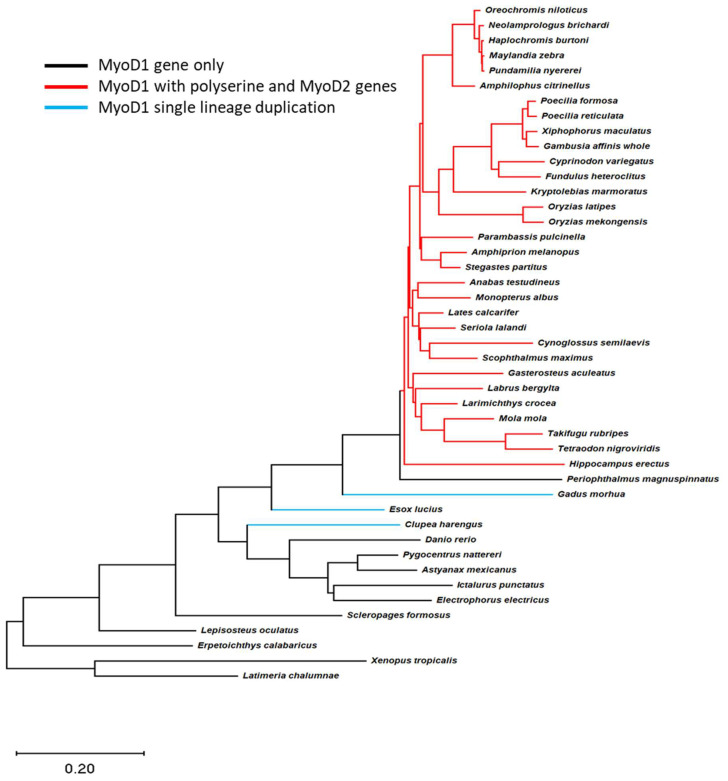
A polyserine insert is found in MyoD1 proteins from species that retain both *MyoD1* and *MyoD2* genes. Teleost species phylogeny is shown for all species with complete *MyoD1* and *MyoD2* data available; the tree by [20] was trimmed using R packages to only include those species with data available. Gene information was subsequently overlaid to this tree and branches coloured to represent the species differences in *MyoD* genes found in the genome. Black lines indicate species with a single *MyoD* gene with no polyserine insert. Red lines indicate the presence of *MyoD1* and *MyoD2* genes where the MyoD1 sequence includes a polyserine insertion. The blue line indicates a lineage specific duplication of *MyoD1*; MyoD1 proteins in this case do not include a polyserine insert. The phylogeny indicates that the polyserine insertion in MyoD1 is only found in species which have retained a *MyoD2* gene.

**Figure 3 jdb-11-00019-f003:**
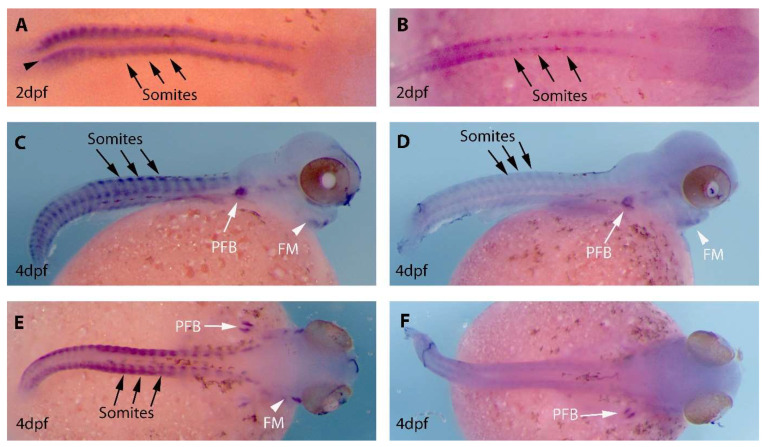
Expression analysis of *MyoD1* and *MyoD2* in the developing embryos of Oreochromis (*Alcolapia) alcalica*. (**A**,**C**,**E**) Lateral and dorsal views of *O. alcalica* embryos analysed by in situ hybridisation for *MyoD1* and (**B**,**D**,**F**) for *MyoD2*. The number of days post fertilisation [dpf] is indicated for each image. Arrows show somites, white arrowheads indicate facial muscle (FM) and white arrows indicate developing pectoral fin bud (PFB). Black dots around the yolk and on the body are chromatophores (pigment cells).

**Figure 4 jdb-11-00019-f004:**
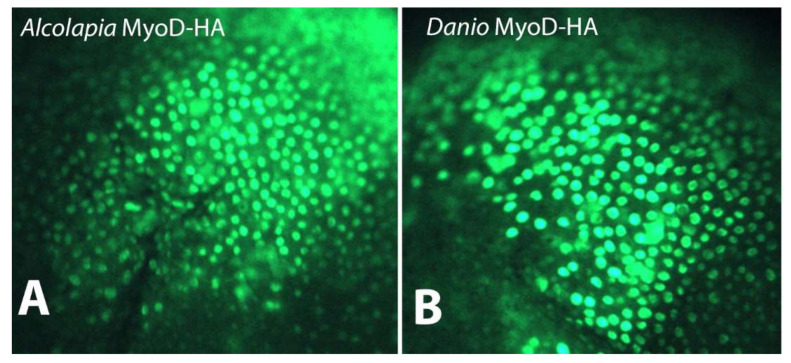
Subcellular localisation of MyoD1 proteins. Immunostaining for HA-tagged MyoD1 proteins from (**A**) *O. alcalica* and (**B**) *D. rerio* shows nuclear localisation when mRNAs coding for these proteins are expressed in *Xenopus laevis* embryos.

**Figure 5 jdb-11-00019-f005:**
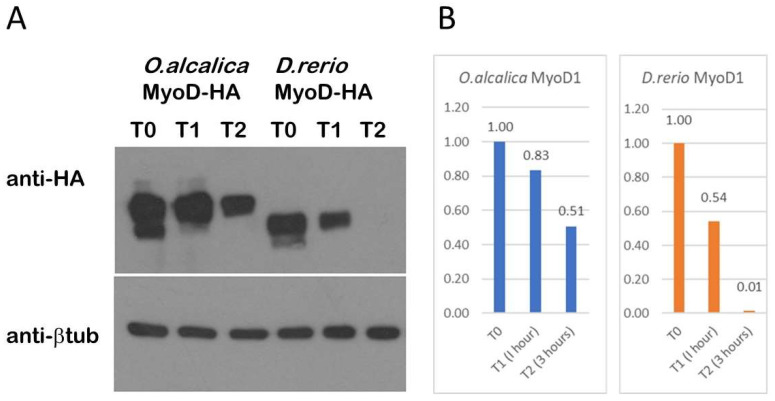
Stability assay shows more perdurance of *O.alcalica* MyoD1 compared to *D. rerio* MyoD1. (**A**) Western blot analysis of extracts from *Xenopus laevis* explants overexpressing HA-tagged MyoD1 proteins. Explants were incubated for either 0 (T0), 1 (T1) or 3 h (T2) in the translational inhibitor, cycloheximide. Anti- beta tubulin is a loading control. (**B**) Quantification of the loss of protein relative to the amount of protein detected at T0 is shown in a graph for *O.alcalica* MyoD1 and for *D. rerio* MyoD1.

**Figure 6 jdb-11-00019-f006:**
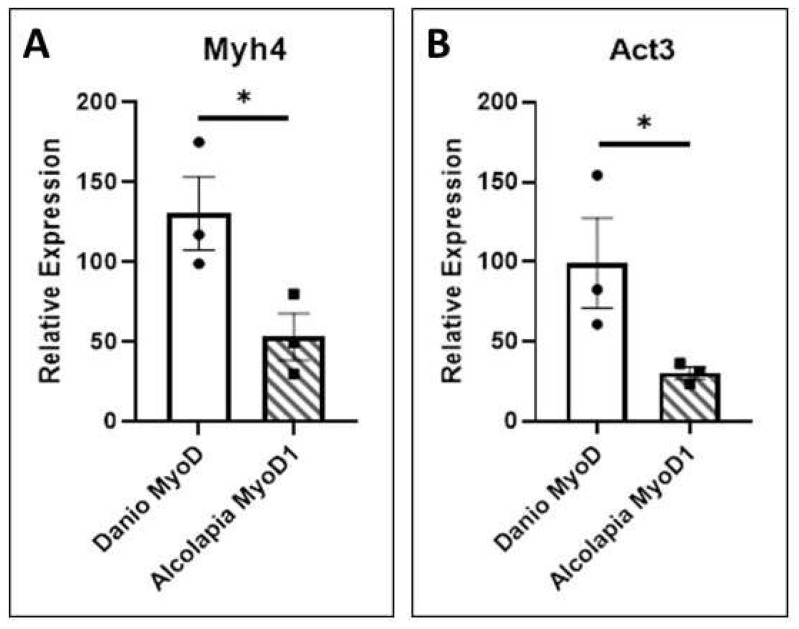
*O. alcalica* MyoD1 and *D. rerio* MyoD1 activate muscle gene expression. (**A**) The expression of *myosin 4* (*myh4*) is quantified by qRT-PCR analysis of RNA extracted from explants expressing *O.alcalica* MyoD1 compared to *D. rerio* MyoD1. (**B**) The expression of *actin 3* (*act3*) is quantified by qRT-PCR analysis of RNA extracted from explants expressing *O.alcalica* MyoD1 compared to *D. rerio* MyoD1. While both MyoD1 proteins can activate high levels of contractile gene expression in naive tissue, there is a significantly higher response to *D. rerio* MyoD1. Experiment was carried out in triplicate and average Ct was normalised to the housekeeping gene *Dicer*. Pairwise *t*-tests comparing the mean relative expression for control and experimental sets for each gene. Error bars represent SEM, * = *p* < 0.05.

## Data Availability

Not applicable.

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
