# Peer review of "The Presence of Two MyoD Genes in a Subset of Acanthopterygii Fish Is Associated with a Polyserine Insert in MyoD1"

_jdb, 2023, doi:10.3390/jdb11020019_

Round 1

Reviewer 1 Report

The manuscript describes the isolation of MyoD genes from the fish Oreochromis (Alcolapia) alcalica and a limited analysis of the genes/proteins. The authors isolated two genes, which they have called MyoD1 and MyoD2. Two MyoD genes have previously been identified in some, but not all, teleost fish. In situ hybridization of MyoD1 and MyoD2  probes on O. alcalica embryos showed that both genes have a similar expression pattern at the two stages chosen.

The protein encoded by MyoD1 has a 22 amino acids Serine rich insert, just N-terminal to the basic domain, and the authors provide evidence that similar inserts are found in other teleost fish with both MyoD1 and MyoD2 proteins. But not in fish with only a single MyoD gene. The authors inject O. alcalica and D. rerio into Xenopus embryos and using immunolocalization show that the Serine-rich insert does not prevent nuclear localisation of  O. alcalica MyoD1. There is only one MyoD protein in D. rerio, which does not have a Serine-rich insert.

The authors looked to see if the Serine-rich insert affected the stability of MyoD1 protein, by injecting mRNA into Xenopus embryos and incubating tissue explants in cycloheximide, a translation inhibitor. Using Western-blots they show that O. alcalica MyoD1 is retained for longer than D. rerio MyoD1, consistent with an increase in stability of the protein.

Finally, using qPCR the authors showed that O. alcalica MyoD1 actvated expression of muscle-specific genes (Act3 and MyH4) when expressed in Xenopus embryos, with D. rerio MyoD1 activating transcription more strongly than O. alcalica MyoD1. 

The results are of good quality and the conclusions appropriate but I have a small number of comments.

1)     Fig 1 shows that in addition to the serine rich insert, there is also an additional 3 amino acid insert withn the basic region. But the authors do not comment about this.

2)     The authors comment that expression of MyoD2 is weaker than that of MyoD1 when determined by whole mount in situ hybridization. Based on this they suggest that “MyoD1 has a higher transcriptional expression in the developing embryo”. While this might be true, WISH is not a true quantitive method and there are other, perhaps less plausible, explanations. If the authors wish to make this claim they should use a more quantitative method, such qPCR.

3)     Why inject mRNA into Xenopus embryos and not zebrafish embryos? I think that the result is correct but perhaps zebrafish would have been a better option. This also applies to the experiments on protein stability and transcription of muscle-specific genes.

4)     For the stabilty of the proteins the authors should at least demonstrate that the cycloheximide has inhibited translation, rather than just assume so. Also a comparison of expression with and without cycloheximide would be an ideal control.

Reviewer 2 Report

The submitted manuscript investigates the evolution, persistence and functional consequences of an unusual polyserine repeat that has arisen in one copy of the developmentally important protein MyoD1 in some teleost fish. Intriguingly, many teleost fish have lost their duplicate gene copy MyoD2, but in all cases (or is it in most cases?) that MyoD2 is retained in evolution then MyoD1 has evolved an extra polyserine region. This raises the question as to what the functional consequences of this repeat is. Or, why have both copies? The authors use a variety of experimental methods to examine expression, nuclear localisation, protein stability and transcriptional activity. There seem to be some differences in the properties of the ‘evolutionarily modified’ MyoD1 protein compared to ‘other’ MyoD/MyoD1 proteins. There are additional experiments that could have been performed, such as comparison to MyoD2 or switching the polySer between proteins, but there is still sufficient of interest and novelty in the paper as it stands.

Minor corrections:

Abstract: “from a number of teleost species” is vague (a number could be one!). Please give number in the abstract.

Methods: “R packages phytools, ape, and Geiger” - reference or github site needs to be given

Latimeria chalumnae needs to be in italics

For the “full-length products” used in riboprobe synthesis, does this mean full length ORF or full length cDNA?

Fig 1. formosa needs lower case f

Figure 1 legend and the text refers to this fish as Oreochromis (Alcolapia) alcalica, but Figure 1 itself calls it Alcolapia alcalica. Consistency needed.

Supplementary Figure S1: Collapsed phylogenetic tree does not add very much information. Is this a collapse of Figure S2? That is not clear in the text or legend. Also, Figure S2 has no legend, what is red in this figure? Does red in Fig S2 mean presence of poly-Ser?

I notice that the ‘gene’ tree in Fig S2 does not correctly recapitulate the likely species phylogeny – this is fine (single gene trees rarely do), but that should be noted in a legend to Fig S2 e.g. something like “The phylogenetic tree shown here, inferred from a single protein alignment, should not be used to infer species relationships. Instead, likely species relationships inferred from previously published studies were used as a framework on which to map presence/absence of the polyserine repeat and the presence/absence of MyoD2”

In the main text, this phrase “This analysis indicates that the inclusion of a polyserine insertion in MyoD1 is strongly linked to the presence of a MyoD2 gene” is ambiguous. What does strongly linked mean? Is it an invariant association? Are there any exceptions?

In Fig S2. Acanthochromis polyacanthus MyoD1 is shown in black. Does this lack the polySer? That would be strange (or interesting) as it is a cichlid so it is in the clade where relatives have polySer. Secondary loss? Annotation error? Incomplete data? Should be commented on.

Line 257: Both Danio and Oreochromis should be in italics.

In the manuscript, Xenopus is sometimes in italics and sometimes not.

The Discussion mentions the lack of correlation between teleost fish WGD and teleost fish diversification, suggesting that WGD could not underpin diversification. This statements overlooks the findings that return to the diploid state after WGD is very slow, such that distinct paralogous genes only evolve long after WGD (Robertson et al. 2017 Genome Biology 18: 111; Parey et al. Genome Research 32: 1685). Hence, the time lag is not an argument against the link between WGD and diversification.

Discussion, italicisation error “is expressed in the adaxial cells” line 376

Reviewer 3 Report

This is a nice study that examines two forms of the muscle specific transcription factor MyoD in a non-model fish species, leading to interesting hypotheses about subspecialization of MyoD1 compared with MyoD2 in species that retain both copies both in terms of localization but also in terms of accrual of a poly-serine repeat in the D1 version.

Major comments/questions:

1)  Figure 1 shows the Alcolapia MyoD1 with the polyserine insertion, in particular in relation to a few other species.  The three species shown with the insertion have similar length and similar distribution of the non-serine amino acids in that region.  While not necessary to expand on this, since the following tree demonstrates a significant number of species with this insertion, are there are any important patterns within the polyserine insertion across the groups that include it?  If it is intrinsicly disordered does the specific location of serine versus non-serine follow any precise pattern or is it a region where there is substantial variation at the individual AA level but a generalized distribution of serines that is the key?

2) For the expression analysis in Figure 3, it is a little challenging to see the adaxial expression of MyoD1 in the 2dpf image, in part as somite development has progressed substantially by that point.  I'm not clear if somite development has essentially reached the tailbud at that point or if there is a significant amount of adaxial tissue beyond the last somite.  It might have been easier to see in an earlier stage embryo. 

3)  While the manuscript mentions differences between MyoD1 and MyoD2 expression in terms of overal amount, there also appears to be differences in localization within somites.  For example at 4dpf, MyoD1 appears to be expressed most strongly in many somites right at the midline.  Is this an enhanced expression in muscle pioneer cells?  It appears that in slightly older somites there is additionally strong MyoD1 expression in the dorsal-most part of the somite.  These patterns seem distinct from patterns in zebrafish, and the images of 4dpf expression of MyoD2 are a little more challenging to see to determine whether perhaps MyoD2 and MyoD1 are expressed in functionally distinct compartments of the somites.

3)  Could a quantification of the protein stability data be included?  Although the manuscript says that D. rerio MyoD1 is expressed robsutly at 1h, it does appear that levels are reduced compared with 0h.  There is also clear reduction in the O alcalica MyoD1 at 1h and 2h, but it could be nice to compare directly the rates of loss of the protein.  The manuscript mentions that Oa MyoD1 could consistently be observed, but it would be nice to know what the variability of expression was in this versus in Dr.

4)  For both the transcriptional data and the protein stability data, were tests done on Oa MyoD2?  Since it would be lacking the poly-serine domain, that might allow a test of whether within this species the form with the polyserine does indeed retain enhanced stability/decreased activity.

5)  On the transcriptional assy, when sequence comparisons of MyoD1 were done at the beginning are there any differences between Oa MyoD1 and Dr MyoD1 (perhaps in particular compared with Xl MyoD1) in the bHLH domain?  Is this assay reflecting differences in transcriptional activation or could it merely be basal DNA binding due to sequence difference in the bHLH domain?

6)  The discussion around line 361 notes that the more stable Oa MyoD1 did not exhibit any enhanced transcriptional activity.  Indeed the results note that it had reduced transcriptional activity.  Does this reflect increased stability but due to proximity with the TAD a real loss of function?

Minor comments/questions:

1)  Latemeria should be italicized in the Methods section.

2)  For the tree shown in Figure 2, I had understood it to be a generalized tree overlaid with information about MyoD duplication and polyserine.  In the discussion (line 327) it mentions alignment of MyoD sequences, and I wasn't sure if that was actually what was presented in Figure 2 or if that was separate data not shown.
